# The Crystal Structure of Mouse Ces2c, a Potential Ortholog of Human CES2, Shows Structural Similarities in Substrate Regulation and Product Release to Human CES1

**DOI:** 10.3390/ijms232113101

**Published:** 2022-10-28

**Authors:** Helgit Eisner, Lina Riegler-Berket, Carlos Francisco Rodriguez Gamez, Theo Sagmeister, Gabriel Chalhoub, Barbara Darnhofer, P J Jazleena, Ruth Birner-Gruenberger, Tea Pavkov-Keller, Guenter Haemmerle, Gabriele Schoiswohl, Monika Oberer

**Affiliations:** 1Institute of Molecular Biosciences, University of Graz, 8010 Graz, Austria; 2Diagnostic and Research Institute of Pathology, Medical University of Graz, 8010 Graz, Austria; 3BioTechMed Graz, 8010 Graz, Austria; 4Institute of Chemical Technologies and Analytics, Faculty of Technical Chemistry, Technische Universität Wien, 1060 Vienna, Austria; 5BioHealth Field of Excellence, University of Graz, 8010 Graz, Austria; 6Institute of Pharmaceutical Sciences, Pharmacology & Toxicology, University of Graz, 8010 Graz, Austria

**Keywords:** carboxylesterase 2c, protein structure, lipid metabolism, X-ray crystallography, α/β-hydrolase fold, biochemistry, lipid hydrolysis, non-alcoholic fatty liver disease, molnupiravir

## Abstract

Members of the carboxylesterase 2 (Ces2/CES2) family have been studied intensively with respect to their hydrolytic function on (pro)drugs, whereas their physiological role in lipid and energy metabolism has been realized only within the last few years. Humans have one CES2 gene which is highly expressed in liver, intestine, and kidney. Interestingly, eight homologous Ces2 (Ces2a to Ces2h) genes exist in mice and the individual roles of the corresponding proteins are incompletely understood. Mouse Ces2c (mCes2c) is suggested as potential ortholog of human CES2. Therefore, we aimed at its structural and biophysical characterization. Here, we present the first crystal structure of mCes2c to 2.12 Å resolution. The overall structure of mCes2c resembles that of the human CES1 (hCES1). The core domain adopts an α/β hydrolase-fold with S230, E347, and H459 forming a catalytic triad. Access to the active site is restricted by the cap, the flexible lid, and the regulatory domain. The conserved gate (M417) and switch (F418) residues might have a function in product release similar as suggested for hCES1. Biophysical characterization confirms that mCes2c is a monomer in solution. Thus, this study broadens our understanding of the mammalian carboxylesterase family and assists in delineating the similarities and differences of the different family members.

## 1. Introduction

Over the last decades, the number of obese individuals has drastically increased and with it the risk for susceptibility to cardiovascular diseases, hypertension, type 2 diabetes, respiratory problems, non-alcoholic fatty liver disease (NAFLD), depression, and cancer [1,2,3]. Furthermore, excess weight can lead to more severe courses of certain diseases as noted, e.g., in obese COVID-19 patients [2]. In adipose tissue, the first step in intracellular lipolysis is primarily carried out by consecutive action of adipose triglyceride lipase (ATGL), hormone sensitive lipase (HSL), and monoacylglycerol lipase [4,5]. Yet studies of Atgl^−/−^- and Hsl^−/−^ mice revealed pronounced residual TG hydrolase activity in non-adipose tissues and members of the carboxylesterase (hCES/mCes) family are suggested to act as additional neutral lipid hydrolases [6,7,8,9,10,11,12]. Enzymes, that play key roles in lipid metabolism and/or metabolic activation of many drugs, like members of the human CES family, are very interesting pharmaceutical targets [10,11,12,13]. The human CES superfamily is divided into five families (CES1, CES2, CES3, CES4A, and CES5A) and all proteins have approx. 560 amino acids [14,15,16]. To date, only human carboxylesterase 1 (hCES1) has been characterized structurally with several three-dimensional (3D) structures in free form and in complex with different substrates and substrate analogs [17,18,19,20,21,22,23]. Based on sequence information and bioinformatic prediction, it is commonly agreed that all CES/Ces family members harbor an α/β-hydrolase fold as their central domain.

Human carboxylesterase 2 (hCES2) is of special interest because of its prominent roles in metabolizing endogenous and exogenous chemicals including many important therapeutic medications, e.g., the anti-cancer chemotherapy drug CPT-11/irinotecan [24,25]. Notably, hCES2 is further described to hydrolytically activate molnupiravir, one of the oral COVID-19 drugs that was recently granted emergency use authorization [26]. Humans have one CES2 gene which is highly expressed in liver, intestine, and kidney [15,27], whereas mice have eight homologous Ces2 genes (Ces2a–Ces2c, Ces2e–Ces2h, and the pseudogene Ces2d). Which one of those represents the ortholog of hCES2 remains controversial, yet the literature reports suggest it to be Ces2c (mCes2c) [6,7,8,11,28]. Obese humans, as well as patients with inflammatory bowel disease, NAFLD, and some types of cancer have lower expression levels of hCES2 compared to healthy individuals leading to lipotoxic accumulation of triacylglycerols in these tissues [6,29]. Similarly, in a murine colitis model and in obese mice, a decline of mCes2 expression is observed in the colon [6,7,29]. Both enzymes, hCES2 and mCes2c, can hydrolyze classical carboxylesterase substrates including acyl-carnitine, they can be inhibited by anti-rat CES antibodies, and they have high activities towards triacylglycerols, diacylglycerols, and monoacylglycerols as substrate [8,11,20,28,29]. However, not all properties of hCES2 overexpression can be mirrored by mCes2c overexpression. Additionally, intensive biochemical analyses of specific hydrolytic activities on neutral lipid substrates show a more complex pattern [6,7,8]. Consequently, Chalhoub et al. very recently proposed that mice do not have the “sole” ortholog of hCES2 with respect to lipid metabolism, but that the activity of hCES2 is carried out by different mCes2 members in a tissue-specific manner in the mouse [8]. Furthermore, based on high sequence similarities and functional overlaps among all mCes/hCES -families, the Lehner group had pointed out the possibility, that the functional ortholog of hCES2 might not come from the murine Ces2 family and suggests that Ces1g is the ortholog of hCES2 [7]. Thus, to translate studies on mCes/hCES family members from model organisms to humans, it is important to establish the role of mCes2 members and identify the orthologs in animal models.

In this work, we take a structure-based approach to better characterize mCes2c and compare its experimental structure with the known structure of hCES1. Our data give detailed insights into the 3D structure of mCes2c determined by X-ray crystallography at high resolution. Despite overall similarities with hCES1, we also noted significant differences regarding binding pockets and the dynamics of the different modules. Furthermore, our in silico docking studies revealed a likely binding pose of the new COVID-19 drug molnupiravir in the substrate binding pocket of mCes2c. Thus, this study starts to fill the large gap in our knowledge of the different murine Ces2 family members and their physiologically relevant interrelationship to human carboxylesterases.

## 2. Results

### 2.1. Purified mCes2c Is Active and Unfolds Irreversibly at a T_M_ of 60 °C

We expressed and purified full-length mCes2c C-terminally fused to a TEV-cleavage site and a 6xHis tag from Expi293F^TM^ cells. The resulting protein has 574 residues and a theoretical molecular mass of 64 kDa. The purified protein was active in hydrolyzing monoolein at a rate of 8900 nmol fatty acids*h^−1^*mg^−1^ in agreement to our recently published data [8] (Figure 1a–c). Mass spectrometry analysis of mCes2c resulted in an experimental mass of 60,897.4 Da corresponding the full-length protein without 21 amino acids at the N-terminus and the 6xHis tag missing at the C-terminus (theoretical molecular mass 60,897.6 Da; mass error 3.6 ppm). In order to assess the overall thermal stability of purified mCes2c, we carried out thermal shift assays which revealed a melting temperature (T_M_) of 58.2 ± 0.5 °C for mCes2c. This high thermal stability was confirmed with CD-spectroscopy by following the changes in molar ellipticity at 222 nm T_M,222_~60.7 ± 0.1 °C (Figure 1d). The thermal unfolding is irreversibly as we observed in the spectra collected at 20 °C upon cooling down from 95 °C.

### 2.2. mCes2c Adopts an α/β-Hydrolase Fold with Residues S230, E347, and H459 Forming a Catalytic Triad

Next, we successfully crystallized purified mCes2c and the crystals diffracted to a resolution of 2.12 Å. The 3D structure was solved using molecular replacement, refined to final R-values of R_work_ = 16.6% and R_free_ = 19.5% (Table 1).

The numbering of mCes2c residues in the crystal structure corresponds to the numbering of the protein in UniProt Entry Q91WG0. The structure contains four monomers in the asymmetric unit (pairwise RMSD values of 0.15 Å), which are referred to as chains A–D, with the biological assembly being a monomer (Figure 2a,b). Amino acids S29-E550 are modeled in chain A, S29-E560 in chain B, E31-E550 in chain C, and P30-L561 in chain D. Electron densities for some loop regions are missing at similar positions in the different chains, namely: G108-K113 in chain A, K51-K54 and N105-S117 in chain C, K51-D52 and L104-M115 in chain D. Missing electron densities in previously mentioned regions are indicative of higher dynamics and disorder. In all 4 chains, residues C97-C125 and C282-C293 form disulfide bonds. The length of the C-terminal helix varies slightly in chains A-D.

The monomeric unit of mCes2c harbors an α/β-hydrolase fold with additional modules and insertions (Figure 2b,c). The core of the α/β-hydrolase domain in mCes2c is built by amino acids V57-M96, L126-L260, D317-G349, and F418-N501, and it contains the active site residues S230, E347, and H459 forming a catalytic triad (Figure 2b–d). The highly conserved oxyanion hole forming region (HGGA, amino acids H149-A152) is part of the substrate binding pocket and is located right after β3 of the α/β-hydrolase core (Figure 2c,d). Compared to the canonical α/β-hydrolase fold, the central β-sheet is extended on both sides, with residues S29-G56 N-terminal of the core domain and P502 to the long C-terminal helix C-terminal of the core (colored gray in Figure 2b,c). Long insertions connecting regions between strands and helices exits as additional modules often observed in α/β-hydrolases (Figure 2b,c) [30]. Since the mCes2c structure revealed high similarity to the experimental structure of hCES1 (PDB 1MX9) we termed these insertions in analogy to the insertions observed in hCES1, namely the “lid” (connecting β1 and β2), the “cap” (connecting β6 and αD) and the so-called “regulatory domain” (RD) connecting β7 with α-helix E (Figure 2b), as described in more detail below [7,20,21,30]. An amino acid sequence alignment of mCes2c, hCES2, and hCES1 is available in the Appendix A.

### 2.3. The Cap, the Regulatory Domain, and a Flexible Lid Region Restrict Access to the Active Site of mCes2c

The cap region (L261-V316, colored teal in Figure 2c,d) contains mostly helices and loop regions and protrudes from the β-strand adjacent to the one forming the nucleophilic elbow (corresponding to β6 in the canonical α/β-hydrolase fold). The so-called regulatory domain (RD, W350-M417) consists of five α-helices and is colored orange in Figure 2b–d. The name originates from its suggested role in hCES1 to regulate the entrance to the active site [17]. The term “lid” was coined to refer to the flexible, partly helical extension after the first canonical β-strand corresponding to amino acids C97-C125 (colored pink in Figure 2b–d). Due to missing electron densities described above, we could not fully model the lid in all chains of mCes2c, especially chains C and D. A short helix (L101-E106) is observed in chain A and partially in chain B and a second shorter helix (L116-S118) is observed in the lid region of chains A and B, whereas no electron density is visible for these helices in chains C and D. The interplay of the cap, the RD and the lid restricts access to the active site of mCes2c.

### 2.4. Substrate Binding Pocket, Active Site, Product Release, and Surface Binding Site of mCes2c

Three ligand binding sites are described in hCES1: the active site, a surface binding site termed Z-site described as low-affinity surface ligand-binding site, and the side door including the gate and switch residues (Figure 3a) [17,20,21].

In mCes2c, the entrance to the substrate binding pocket of each monomer is framed mostly by flexible loops and helical parts of the cap, the RD including the gate residue M417, and parts of the core. The gate residue is highly conserved between hCES1, hCES2, and mCes2c, and is positioned right at the end of the RD (Figure 2d and Figure 3a). Mainly hydrophobic amino acids surround the potential entrance to the substrate binding pocket, namely A152, L261, P262, D263, M311, I352, M356, and M417 (Figure 3c). The substrate binding pocket itself is also predominantly lined by hydrophobic residues, namely M103-P110, M112-M114, L116, W147, H149-G156, M160, F161, E229-A231, L264, F348, W350-V356, F418, Y446, F447, R451, D458-D461, I463, P464, F470-W472, and M474 (Figure 3d). In each monomer of the experimental structure of mCes2c, additional electron density was observed that could ideally be fitted to one molecule of nicotinamide (NCA) in the substrate binding pockets with 100% occupancy (Appendix A). Analysis of the ligand–protein interaction reveals H-bonding of the NCA-carbonyl oxygen to the backbone amide protons of oxyanion-hole residues G151, A152, and A231. Additionally, H-bonding to a modelled water molecule is observed (Appendix A). The presence of NCA leads to restricted access to the active site within the core (Figure 3c,d). Very likely, the NCA molecule is derived from the culture medium and was co-purified with mCes2c.

Similar to hCES1, we also observed a small Z-site, described as low-affinity surface ligand-binding site, which is responsible for shifting the trimer-hexamer equilibrium in hCES1 [21]. The Z-site is framed by three α-helices of the RD and located directly next the active site. The catalytic triad residue E347 is on a turn after β7 just prior the first helix of the RD that separates the active site and Z-site region. Amino acid residues involved in forming the Z-site are: G349, W350, P353, V354, A359, I362, K363, Q403, Q406, I407, F409, T410, P452, and H454. In the structure we found electron density at the Z-site and could fit a single NCA molecule in chains A and B. Only residual electron density can be observed in chains C and D, which could not be fitted unambiguously to a distinct small molecule with full occupancy (Figure 3b, Appendix A).

A second entrance to the active site called side door has been suggested for rabbit liver carboxylesterase (rCE) and hCES1 on the basis of binding of the anti-cancer drug CPT-11 in rCE and the fatty acid palmitate in hCES1 [21,31]. Two cavities in mCes2c are in a similar position as the side door in hCES1 (Figure 3e,f). Both cavities are lined with mainly hydrophobic residues. Side door cavity 1 is embedded by the long C-terminal helix, a kinked helix connecting the RD with the core, and a loop region from the cap. Side door cavity 2 is framed by a helix in the region of N369-M381 from the RD and by a helix from the cap (S269-I302). Side door cavity 2 is directly connected with the substrate binding pocket. The connection is centered between residues W350 and residue S358 of the RD, the region around L261 and V308 of the cap, and by the kinked helix harboring the gate residue M417 and the switch residue F418 (Figure 3f,g). This kinked helix is a part of the product release, the Z-site and the substrate binding pocket. A similar helix is also observed in hCES1. Structurally, these helices are conserved (RMSD of 0.24 Å over 17 from 17 residues) between hCES1 and mCes2c despite a low sequence identity of 25.8% (Figure 3g).

### 2.5. SAXS, SEC, and SEC MALS Demonstrate That mCes2c Is Monomeric in Solution

To determine the oligomerization state of mCes2c we performed independent biophysical techniques, including SEC, SEC-MALS and SAXS. In the SEC experiment, mCes2c (c = 1.5 mg/mL) showed a distinct peak at an elution volume of 15 mL. Based on our standard run, this corresponds to approx. 48 kDa, indicating that mCes2c is monomeric in solution (Figure 4a). An additional, minor peak at an elution volume of 13.2 mL corresponds to 108 kDa, indicating a minor population of dimeric mCes2c. The SEC run was also coupled to MALS detection. The two analyzed peak areas were monodisperse ($\;\overset{\text{¯}}{M}$w/$\;\overset{\text{¯}}{M}$n < 1.01) and the estimated molar masses were 54 kDa for the monomeric peak and 108 kDa for the dimeric peak (Figure 4b). The monomeric oligomerization state of mCes2c was corroborated by SAXS data from which we calculated a radius of gyration of 24.6 Å, a maximum diameter of 83.0 Å, and a calculated molecular weight of 50.8 kDa. The low-resolution shape of mCes2c reconstructed from SAXS profiles fit (ꭓ^2^ = 1.4) to the experimentally determined crystal structure of mCes2c (Figure 4c,d). Together the biochemical, structural, and biophysical experiments all corroborate the predominantly monomeric state of mCes2c.

### 2.6. Elevated B-Factors Indicate Higher Flexibility and Thermal Motion of the RD and the Lid Region

Next, we analyzed the B-factors as measure of atomic mean squared displacement of different domains and modules of mCes2c. The majority of the core domain does not show high B-factors and thus appears rather rigid as expected. Interestingly, even the cap region (teal, Figure 5) of mCes2c including the loop region V308-V316 appears not to be very flexible (average B-factor of 25.2 Å^2^). However, a loop in the lid region (N100-L116) (pink, Figure 5) exhibits high flexibility (B-factors ranging between 21.1–128.9 Å^2^, average of 64.5 Å^2^). This is in agreement with missing electron densities due to disorder for some of these residues in different chains of the lid. Furthermore, the RD shows higher values for the B-factors (average B-factor 46.1 Å^2^) which indicate flexibility and might enable breathing motions of mCes2c. Hence, entrance to the substrate binding pocket might be mediated by the flexible lid region and increased dynamics in the RD (Figure 5).

### 2.7. In Silico Docking Studies Show a Possible Binding Pose of the COVID-19 Prodrug Molnupiravir in the Active Site of mCes2c

The prodrug molnupiravir (EIDD-2801) was docked into mCes2c by using Maestro, Schrödinger. As a result, molnupiravir is positioned in the active site pocket of mCes2c with an extra precision (XP) glide score of −6.3 (Figure 6a). The majority of the molecule is embedded in a hydrophobic pocket of the substrate binding site, additionally stabilized by H459, S351, and a H-bond to the backbone of A107. Remarkably, the catalytic S230 is in a 3.4 Å distance from the carbonyl carbon of molnupiravir to potentially facilitate the initial nucleophilic attack for the hydrolytic reaction (Figure 6b). Therefore, we speculate that mCes2c enables the prodrug bioactivation of molnupiravir to β-D-N4-hydroxycytidine which is then further transformed to the active drug.

## 3. Discussion

### 3.1. Differences in the Entrance to the Substrate Binding Pocket of mCes2c Compared to hCES1

Recently, mCes2/hCES2 family members were investigated intensively with respect to their ability to hydrolyze lipids, and their possible role in lipid signaling pathways [6,7,8]. Mammalian carboxylesterases are localized within the lumen of the endoplasmic reticulum in many tissues [14,24]. mCes2c is mainly expressed in the colon but is also present in the duodenum, the ileum, the jejunum, and the liver. Thus, mCes2c partially shows an expression pattern similar to that of hCES2, which is expressed in the small intestine, kidney and liver, whereas hCES1 expression occurs mainly in the liver and lung [6,32,33].

In this work, we determined the crystal structure of mCes2c (561 amino acids) which is suggested to be the mouse ortholog of hCES2 (559 amino acids, Appendix A). The sequence identity of hCES2 and mCes2c amounts to 71.2%, the sequence similarity is 88.7%. Among all mCes2 proteins, mCes2c shows the highest sequence similarity to hCES2 which is also depicted in a similar pattern of substrate specificity against TGs, DGs, and MGs [8]. hCES1 has 567 amino acids and shares 44.3% sequence identity and 71.6% sequence similarity to mCes2c. hCES1 mainly hydrolyzes substrates with a small alcohol group and a large acyl group whereas hCES2 hydrolyzes substrates with a large alcohol group and a small acyl group [7].

Since the experimental structure of hCES2 has not yet been determined, we compare mCes2c with the known 3D structure of hCES1. With respect to the overall structure, mCes2c and hCES1 share pronounced similarities, namely the α/β-hydrolase core domain and the extended β-sheet with three β-strands N-terminal and two β-strands C-terminal of the α/β hydrolase core. Superposition of these two carboxylesterases shows very high structural similarity with an RMSD of 0.58 Å over 367 Cα-atoms being aligned. Regardless of the similarities of mCes2c and hCES1, the experimentally determined 3D structures and the monomeric structural units of hCES1 and mCes2c exhibits significant differences. These include regions likely involved in substrate entry, namely in the lid, cap, and a loop region within the α/β-core domain (Figure 7).

A prominent difference between mCes2c and hCES1 is the loop region in the cap, which is twelve amino acids longer based on the structural alignment (K302-V322) and is responsible for the closed entrance to the substrate binding pocket in hCES1 (Figure 7a,b). Additionally, this loop region in hCES1 shows higher values for the B-factors which indicate flexibility (average number of B-factor 60.6 Å^2^, Figure 7b). In mCes2c this loop is shorter (V308-V316) and the whole cap region is—based on the B-factors—not flexible (average B-factor 26.2 Å^2^). It is possible that flexibility in this region is responsible for the opening-closing mechanism of the substrate binding pocket of hCES1 together with the flexible regions of the RD domain of hCES1 (Y366-S380: average B-factor 74.0 Å^2^; K393-L416: average B-factor 71.1 Å^2^). The entrance to the substrate binding pocket of mCes2c might be mediated by a flexible loop in the lid region (N100-L116) and dynamics in the RD (based on B-factors, see Section 2.6). mCes2c also differs significantly in the flexible lid region (N100-L116, indicated in pink) with high flexibility (average B-factor 64.5 Å^2^; Figure 2c) and a loop region of the core domain (Figure 2d).

In hCES1 the substrate binding pocket opens on its top (between cap and RD), while in mCes2c the pocket opens on the side (between lid and RD), with closer proximity to the active site compared to hCES1. Whether different substrates enter via different entrances in carboxylesterases needs to be established in further structural studies with different substrates and different mCes/hCES family members.

### 3.2. Similarities in the Product Release between mCes2c and hCES1

For hCES1, three features are described for the side door. Bencharit et al., 2006 describes the highly conserved gate residue M425 in hCES1 as the most important residue for regulating the release of products such as fatty acids. The highly conserved switch residue F426 might be a link between RD and the catalytic region, allowing a rotation of RD. The aromatic releasing valve residue F551 of hCES1 is the third feature and might be responsible for long-chain fatty acid release [21]. Similar to the product release mechanism of hCES1 proposed by Bencharit et al. next to the active site in the site door region, mCes2c harbors the conserved gate (M417) and switch (F418) residues (Figure 3a). However, there is no aromatic residue in mCes2c equivalent to the releasing valve residue (F551 in hCES1) in the terminal helix which has been observed in association with the product [21].

### 3.3. The Oligomerization State of mCes2c Is Different to hCES1

Bencharit et al. describe a regulatory mechanism for hCES1 that is dependent on its oligomeric state [21]. As hexamer, the proteins are arranged in a way that the active site is not accessible for the substrate. Upon binding of a ligand or a substrate to the Z-site, the hexamer cannot form, and the equilibrium is shifted towards a trimer (called trimer-hexamer equilibrium). All of our independent experimental approaches point toward a primarily monomeric form of mCes2c in absence of substrates. In addition, the arrangement of the crystal packing (four molecules in the asymmetric unit) does not indicate to form the interface Z-site on the regulatory domain that is suggested to be important for the oligomerization and regulation in the hCES1. This highlights an additional difference to human CES1. Interestingly, the Z-site is a surface binding site which is conserved between hCES1 and mCes2c and is definitely capable to bind ligands as indicated in chain A and chain B of mCes2c by NCA binding (Figure 3b). It has been speculated that the high-mannose N-linked glycosylation of hCES1 plays a role in trimer stabilization [20,21], whereas other reports have described no significant effects of the N79 glycosylation for hCES1 [18]. No glycosylation is observed in the structure of mCes2c. Additionally, no hints for glycosylation are given by mass spectroscopy which is in full agreement with the first literature reports on the enzyme [28].

### 3.4. hCES2/mCes2c and Their Ability to Activate the Antiviral Prodrug Molnupiravir

Molnupiravir is a novel broad-spectrum antiviral agent that acts by introducing mutations into viral RNA via RNA-dependent RNA polymerase. Although initially developed against influenza, it obtained world-wide attention as treatment strategy for COVID-19 using the brand name “Lagevrio^®^”. In vitro and in vivo studies of molnupiravir showed high efficacy to the replication machinery, high genetic barrier to resistance, and cytotoxicity. Molnupiravir is the isopropylester of the ribonucleoside analog β-D-N4-hydroxycytidine. It is administered orally and first needs to be hydrolyzed to N-hydroxycytidine, and then phosphorylated before it becomes incorporated in place of cytidine triphosphate or uridine triphosphate during RNA synthesis. Subsequently, these erroneous incorporations lead to numerous mutations in the whole viral genome, ultimately rendering the virus unable to replicate [34,35,36,37]. Recent studies demonstrates that recombinant hCES2—yet not hCES1—hydrolyzes molnupiravir, whereas hCES1 hydrolyzes the larger ribonucleoside analog prodrug remdesivir much more efficient than hCES2 [26,38]. Our in silico docking studies of the experimental structure of mCes2c with molnupiravir indicate an orientation of the prodrug that might enable prodrug activation. It will also be of high interest in the future to compare experimental structures of mCes2c in complex with molnupiravir and hCES1 complexed with remdesivir. The differential hydrolytic abilities are also of potential pharmacological consequences since mCes2/hCES2 and hCES1 vary in their protein expression levels in different tissues. On the long run, a combination of structure–activity studies of potential prodrugs for mCes/hCES family members might be envisaged along with different administration schemes depending on tissue-specific expression patterns.

## 4. Materials and Methods

### 4.1. Nomenclature

In this manuscript, we use the suggested nomenclature of Holmes et al., 2010 when referring to human and mouse carboxylases, namely capitalized CES1-CES2 for proteins from humans and Ces2a-Ces2h for mouse proteins [16]. For ease of reading and discussion purposes, lower case letters ‘h’ or ‘m’ additionally indicate human or mouse proteins, respectively.

### 4.2. Materials

Chemicals were obtained from Sigma-Aldrich (St. Louis, MO, USA) or Carl Roth GmbH (Karlsruhe, Germany), if not stated otherwise; columns for protein purification were obtained from Cytiva (Uppsala, Sweden). PierceTM Unstrained Protein MW Marker from Thermo Fisher Scientific was used as size marker for SDS-PAGE gels, unless otherwise noted. For protein production we used the mammalian Expi293F™ cells from Thermo Fisher Scientific.

### 4.3. Cloning, Expression and Purification of mCes2c

The cloning of mouse Ces2c (mCes2c, amino acids M1-L561) into a pFLAG CMV 5.1 expression vector was described previously [8]. The final construct encodes for mCes2c according to UniProt Q91WG0, harboring an N-terminal endoplasmic reticulum (ER) localization signal (M1-D28) and a C-terminal ER retention signal (HREL) followed by a C-terminal TEV-cleavage site and a His_6_-tag. For protein expression, the resulting vector was transfected into Expi293F™ cells using the ExpiFectamin 293 transfection kit. Seventy-two hours after transfection, we collected the medium containing the overexpressed and secreted protein, centrifuged, and then filtered it through a 0.45 µm filter to remove cell debris. Afterwards, we purified His_6_-tagged mCes2c protein by Ni-affinity chromatography followed by size exclusion chromatography (SEC) using a Superdex 200 Increase 10/300 GL column buffer A containing 20 mM Tris/HCl, pH 7.4, and 150 mM NaCl. The concentration of the purified protein was determined via absorption at 280 nm with the calculated extinction coefficient 73,130 M^−1^ cm^−1^; the purity was analyzed using SDS-PAGE.

### 4.4. Biophysical and Biochemical Characterization of mCes2c

#### 4.4.1. Activity Assay

The activity assay was performed using monoolein as substrate as described in Chalhoub et al. [8].

#### 4.4.2. Protein Characterization Using Size Exclusion Chromatography (SEC) and Multi-Angle Light Scattering (MALS)

For characterization by SEC-MALS, purified mCes2c (1.5 mg/mL) was centrifuged at 12,851× *g* for 15 min at 4 °C prior the actual experiment. mCes2c was eluted from a Superdex 200 Increase 10/300 GL column (GE Healthcare) in a buffer B containing 20 mM potassium phosphate, pH 7.4 and 150 mM NaCl with a flow rate of 0.3 mL/min at 8 °C directly coupled to a MALS unit from Wyatt Technologies (TREOS II) equipped with three detection angles. Data were analyzed with the program ASTRA 7.1.2.5 (Wyatt Technologies). To determine the molecular weight based on the SEC, protein standard from Bio-Rad (Gel filtration Standard, 1511901) was used (equation for linear regression: y = −0.1963x + 4.6251; R^2^ = 0.9984).

#### 4.4.3. Thermostability—Differential Scanning Fluorimetry

For differential scanning fluorimetry, we mixed protein and dye solution in 96 well plates to a total volume of 20 µL per well. 4 µL mCes2c per well (1.3 mg/mL) were mixed with 14 µL buffer B, 2 µL of 50X SYPRO Orange (Thermo Fisher Scientific) per well were added to reach the final concentration of 5X for SYPRO Orange. The temperature was increased from 20 °C to 95 °C with increments of 0.5 °C every 30 s. The melting temperature (T_M_) was calculated from the first derivate of the fluorescence signal vs. the temperature. Experiments were performed as triplicates.

#### 4.4.4. Thermostability—Circular Dichroism Spectroscopy

A temperature interval scan measurement of mCes2c (0.2 mg/mL in buffer B) was performed at 222 nm every 0.5 °C/min with a JASCO J-1500 CD spectrophotometer using a 1 mm path length quartz cuvette. Spectra were measured at six accumulations collected at 100 nm/min scanning speed, data pitch of 0.2 nm, a digital integration time (DIT) of 4 s and a bandwidth of 2 nm.

#### 4.4.5. Mass-Spectrometry

Purified mCes2c was desalted using 10 kDa Amicon Ultra 0.5 mL centrifugal filter units (Millipore, Billerica, MA, USA). A final protein concentration of 200 ng/µL in 2% acetonitrile and 0.1% formic acid (in water) was obtained. Protein species were separated by nano-HPLC (Dionex Ultimate 3000) equipped with a Pepswift precolumn (monolithic, 5 × 0.2 mm) and a ProSwift RP-4H column (monolithic, 100 µm × 25 cm; all Thermo Fisher Scientific, Vienna, Austria). Two µL of protein sample were injected and concentrated on the enrichment column for 3 min at a flow rate of 5 µL/min with 0.1% formic acid as isocratic solvent. Separation was carried out on the nanocolumn at a flow rate of 1 µL/min at 37 °C using the following gradient, where solvent A was 0.1% formic acid in water and solvent B was acetonitrile containing 0.1% formic acid: 0–2 min: 5% B; 2–17 min: 5–60% B; 17–20 min: 60% B; 20–20.1 min: 60–5% B; and 20.1–29 min: 5% B. The maXis II ETD mass spectrometer (Bruker, Bremen, Germany) was operated with the captive spray source in positive mode with following settings: mass range: 300–3000 *m*/*z*, 1 Hz, source voltage 1.3 kV, dry gas flow 3 L/min at 180 °C. The protein mass spectra were deconvoluted by Bruker Data analysis software, using the MaxEnt2 algorithm. The following main parameters were applied: *m*/*z* range, min. 5000 to max. 100,000, instrument resolving power was set to 10,000. For peak detection, the Sum-peak method was used.

### 4.5. Structural Characterization

#### 4.5.1. Small-Angle X-ray Scattering (SAXS) Measurements

SAXS measurements were performed at the BM29 beamline for Bio-SAXS at the European Synchrotron Radiation Facility (ESRF, Grenoble, France). Three samples of purified mCes2c with a volume of 50 µL were prepared with three different concentrations (2.5, 5.0 and 10 mg/mL). Samples and buffer A were shipped to ESRF Grenoble and the measurement was performed on site. The sample delivery and measurements were performed using a 1 mm thick quartz capillary, which is a part of BioSAXS automated sample changer unit. Before and after each sample measurement, the corresponding buffer was measured and averaged. Low-q regions of the buffer subtracted averaged frames were studied to identify aggregation. The Guinier approximation was used to extract the radius of gyration (*Rg*) from the scattering data. All the experiments were conducted with the following parameters: beam current = 200 mA, flux = 2.6 × 10^12^ photons/sec, wavelength = 1 Å, estimated beam size 1 mm × 100 µm, 10 frames (1 frame per second) were taken from each sample. All experiments were standardized to water (100% transmission, 20 °C. Scattered intensities were recorded using a Pilatus3 2M in-vac detector. Data analysis was performed using the ATSAS 3.1.1 software (EMBL, Hamburg, Germany) suite [39].

#### 4.5.2. Crystallization, Data Processing, and Refinement

The monomeric peak fractions from the final SEC run in buffer A were concentrated to 2.7 mg/mL and different standard crystallization screens were used for setting up the crystal plates. Crystallization experiments were performed with an ORYX 8 robot (Douglas Instruments, Hungerford, UK) using the sitting drop vapor-diffusion method in 96-well plates. The drops contained equal volumes (0.5 µL) of protein and reservoir solution and were equilibrated against 30 µL of reservoir solution. Crystallization condition containing 0.2 M ammonium sulfate, 0.1 M sodium cacodylate buffer pH 6.5 and 30% *w*/*v* PEG 8000 (ShotGun (SG1) Screen C5, Molecular Dimensions, Newmarket, UK). We obtained crystals of ~200 μm after 1 month of incubation at 20 °C. A complete diffraction dataset of mCes2c up to 2.12 Å, space group P2_1_ 2_1_ 2_1_, resolution was collected at the DESY synchrotron facility PETRA III, P11 beamline in Hamburg, Germany. Data were processed using XDS and scaled using Aimless [40,41,42]. The phase problem was solved by molecular replacement using human CES1 (PDB-ID 1MX9, 44.3% sequence identity to mCes2c) as template. The structure was built using Coot [43] and Isolde [44] and refined with REFMAC5 within the CCP4-Software Suite [41,42,43]. Waters were placed by using the update waters option in Phenix [45]. The refined X-ray models were validated by MolProbity and with the PDB validation report [46]. The residues M1-D28 of mCes2c were not observed in the electron density and omitted from the final model. Detailed data processing and structure refinement statistics are summarized in Table 1. All structure-related figures and structure alignments were generated using PyMOL (The PyMOL Molecular Graphics System, Version 4.6 Schrödinger, LLC). Cavities were calculated with the PyMOL plugin CavMan (Innophore GmbH, Graz, Austria). RMSD values were calculated using the pymol-command ‘align’. The atomic coordinates and structure factors have been deposited in the Protein Data Bank under the accession code PDB ID 8AXC.

### 4.6. In Silico Docking Studies with the Novel COVID-19 Prodrug Molnupiravir

Maestro (version 12.6.144), Schrödinger (2020-4) was used for the in silico docking analysis [47]. The structure of molnupiravir (EIDD-2801) was adapted from Pubchem [48] and drawn using the 2D sketcher tool of Maestro. It was then proceeded by the ligand preparation with the Ligprep tool. For protein preparation, chain B of mCes2c was processed by removing the water molecules, adding hydrogen atoms, and then performing the restrained minimization with the protein preparation wizard. This was followed by generating a 25 Å receptor grid with S230 as center. Finally using the ligand docking tool, the prepared ligand was docked into the generated grid of mCes2c. The Extra-precision (XP) glide docking method was employed with a rigid ligand sampling parameter [49]. The docking result was analyzed by viewing the ligand interaction diagram, the docking scores, and using the PyMOL suite (v 2.0 Schrödinger, LLC, New York, NY, USA).

## Figures and Tables

**Figure 1 ijms-23-13101-f001:**
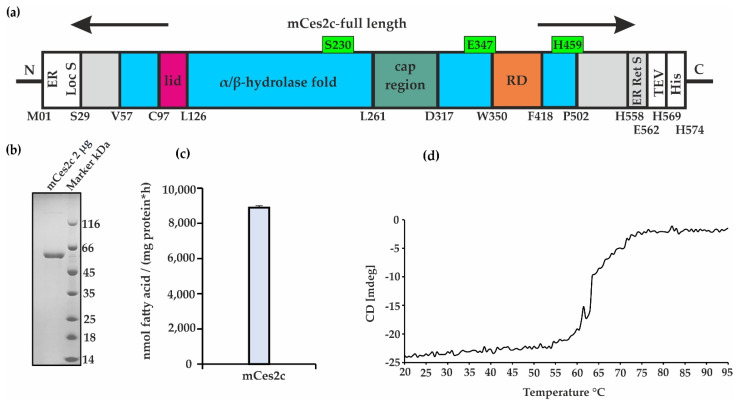
**mCes2c expression, purification, hydrolytic activity, and thermal stability.** (**a**) Scheme of mCes2c used in this study. The lid is shown in pink, the α/β-hydrolase core is indicated in blue, the cap region in teal, and the regulatory domain (RD) in orange. Catalytic triad residues (S230, E347, H459) are indicated in green. The gray parts indicate N- and C-terminal extensions of the classic α/β-hydrolase core and the endoplasmic reticulum retention signal (ER Ret S) visible in the crystal structure (PDB 8AXC). The white parts represent the endoplasmic reticulum localization signal (ER Loc S), the TEV-cleavage site, and the C-terminal 6xHis tag. (**b**) SDS-PAGE of purified mCes2c. (**c**) Neutral lipid hydrolase activity assays were performed with purified mCes2c upon incubation with monoolein. The released fatty acids were monitored as readout of hydrolysis. (**d**) Thermal unfolding of mCes2c monitored with CD spectroscopy.

**Figure 2 ijms-23-13101-f002:**
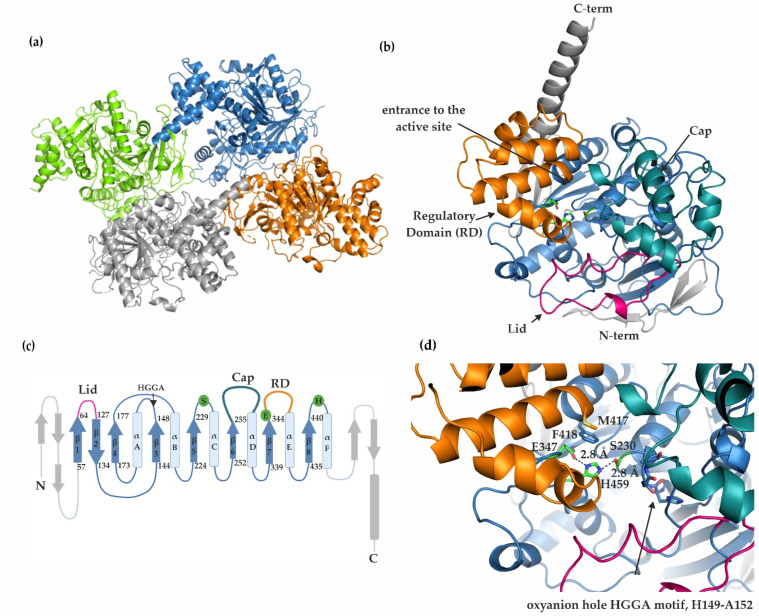
**Overall structure of mCes2c.** (**a**) The asymmetric unit includes 4 molecules, chain A is shown in green, chain B in blue, chain C in orange, and chain D in gray. (**b**) mCes2c in cartoon representation. The structure shows a conserved α/β-hydrolase fold typically for mammalian carboxylesterases (shown in blue). The catalytic triad residues S230, E347, and H459 are represented as green sticks. The lid is depicted in pink, the cap in teal and the regulatory domain (RD) in orange. Distinct sites as discussed in the paper are indicated with arrows. (**c**) Topology diagram of mCes2c. mCes2c harbors an α/β-hydrolase fold with eight mostly parallel β-strands, whereby only the second strand is antiparallel; the strands are connected by six α-helices. The positions of the catalytic residues in the loops connecting strands and helices are conserved, namely: the nucleophilic serine after strand β5, the acidic residue positioned after strand β7, and the conserved histidine after strand β8. Please note that secondary structural elements are not included for the additional modules such as the lid, the cap, and the RD; loop regions are not to scale. (**d**) The active site residues S230, E347, and H459 are indicated as green sticks. Distances of the H-bonding network and the oxyanion hole motif are indicated. The gate M417(orange) and switch F418 (blue) residues are shown in sticks and are highly conserved among hCES1, hCES2, and mCes2c (see also Appendix A).

**Figure 3 ijms-23-13101-f003:**
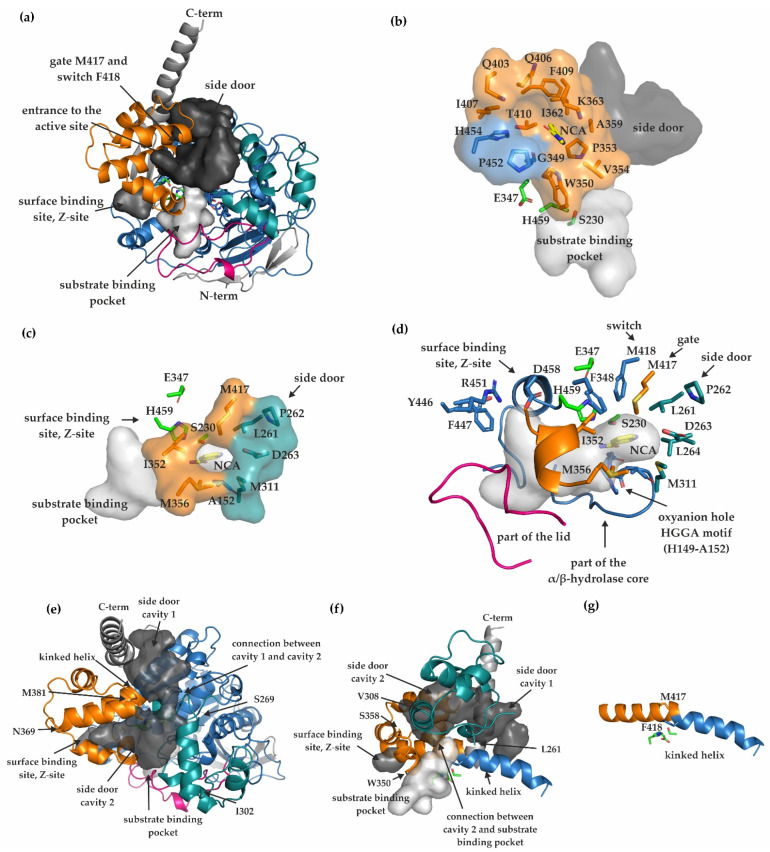
**Binding pockets and conserved sites of mCes2c.** (**a**) Substrate binding pocket (light gray), side door (dark gray) and surface binding site (Z-site, gray) are formed by mCes2c. The structure is shown in the same orientation as in Figure 2b. (**b**) Surface binding site (Z-site) of mCes2c (chain B) with residues sculpting the site colored according to their underlying modules (RD in orange, cap in teal, α/β-hydrolase core in blue). The active site residues are green and the nicotinamide (NCA) is in yellow. The molecule is oriented upon an approx. 90 degrees relative of 3a. (**c**) The entrance to the substrate binding pocket in surface representation is colored according to the underlying modules; residues lining the entrance are in sticks representation, A152 of the α/β-hydrolase core is in blue. (**d**) Substrate binding pocket (light gray surface) with residues lining the entrance in sticks. Other residues which form the substrate binding pocket are shown as sticks and cartoon, respectively (lid in pink, part of the RD shown in orange, and parts from the core are in blue). The NCA is yellow. The orientation of mCes2c is equivalent to Figure 3c. (**e**) Positioning of the side door (dark gray, surface representation) of mCes2c; RD in orange, cap in teal. (**f**) Positioning of the side door (dark gray, surface representation) and the connection between side door cavity 2 and the substrate binding pocket (light gray, surface representation). The kinked helix is indicated with an arrow. (**g**) The kinked helix represented as cartoon harbors the gate and switch residues M417 and F418, respectively. The catalytic residues (S230, E347, H459) are indicated as green sticks. The helix is presented in the same orientation as in Figure 3f.

**Figure 4 ijms-23-13101-f004:**
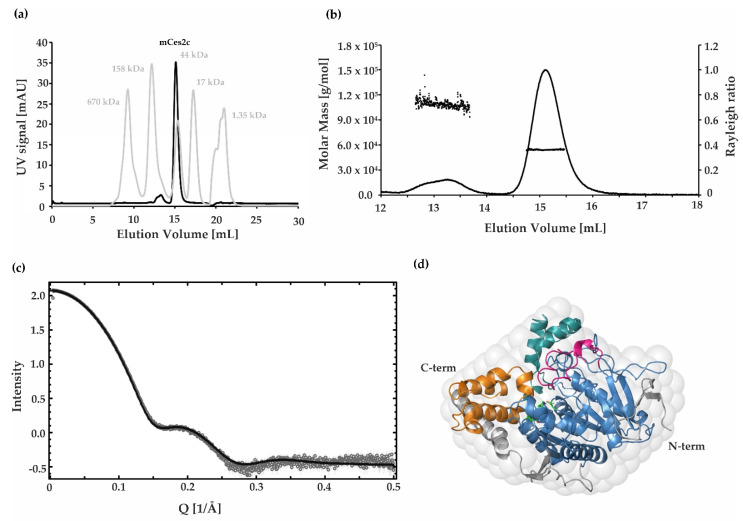
**mCes2c is a monomer in solution.** (**a**) mCes2c was analyzed using a Superdex 200 Increase analytical SEC column. The UV signal was recorded at 280 nm. Gray peaks show the elution volumes of the protein gel filtration standard from BioRad (670 kDa: thyroglobulin; 158 kDa: γ-globulin; 44 kDa: ovalbumin; 17 kDa: myoglobin; 1.35 kDa: vitamin B12). (**b**) The weight average molar mass (MW of 54.0 ± 2.4 kDa) of the dominant peak was determined by multi angle light scattering (MALS) ± standard error of mean (SEM). The very small fraction eluting at 13.2 mL likely corresponds to a dimeric form of mCes2c. (**c**) SAXS scattering curve of purified mCes2c indicates a globular monomeric protein in solution. Dmax was calculated as 83.0 Å with a radius of gyration of 24.6 Å. (**d**) A low resolution ab-initio model (gray surface) derived from the SAXS scattering curve fits very nicely to the experimentally determined 3D structure of monomeric mCes2c. All calculations and ab-initio models are obtained using the ATSAS 3.0 software.

**Figure 5 ijms-23-13101-f005:**
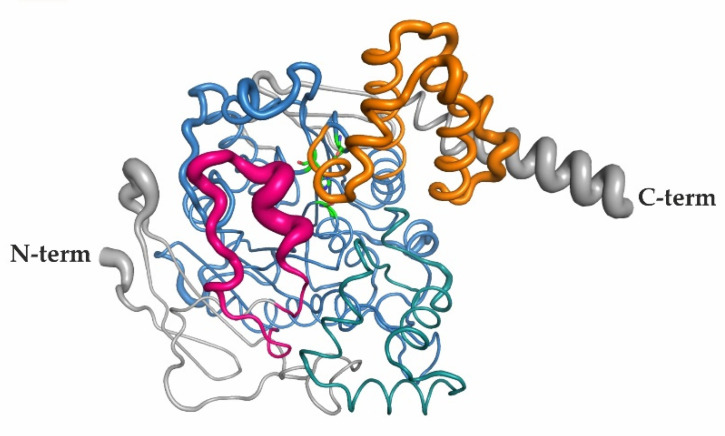
**Overall structure of mCes2c in tube representation.** The flexibility of individual residues indicated based on B-factor values. The cap residues (teal) have an average B-factor of 25.2 Å^2^, whereas the lid region (N100-L116, indicated in pink) exerts high flexibility resulting in an average B-factor of 64.5 Å^2^. The α/β-hydrolase core is shown in blue, RD in orange, catalytic triad residues S230, E347, and H459 are shown as green sticks.

**Figure 6 ijms-23-13101-f006:**
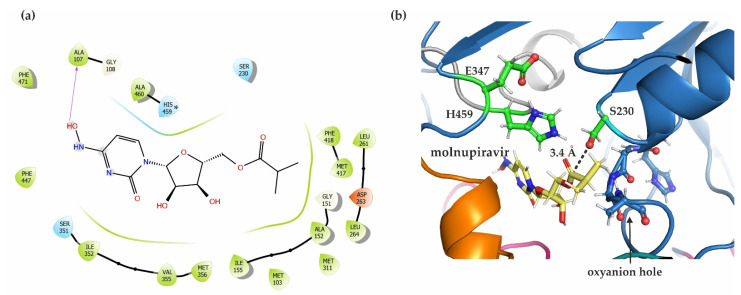
Molecular docking of the prodrug molnupiravir into mCes2c. (**a**) The ligand interaction diagram shows mCes2c residues within a 3.5 Å radius of molnupiravir. Pink arrow denotes H-bonding interaction. The asterisk * is placed to indicate that H459 is in the doubly protonated state. (**b**) 3D image displaying the proximity of active site residues to the ligand. The distance from carbonyl carbon of molnupiravir and the oxygen of S230 is 3.4 Å. The catalytic triad and oxyanion hole residues are depicted as green and blue sticks, respectively.

**Figure 7 ijms-23-13101-f007:**
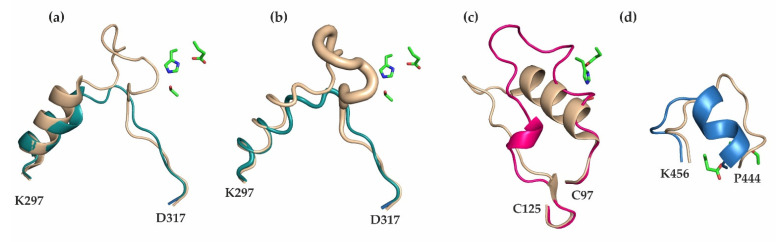
**Comparison of mCes2c with hCES1**. (**a**) The cap of hCES1 (wheat) has an extra insertion in a loop compared to mCes2c (teal). (**b**) B-factors in tube representation indicate lower flexibility of the mCes2c cap compared to that of hCES1. (**c**) Differences of mCes2c (lid region in pink) and hCES1 (wheat) in the lid region. (**d**) A loop region from the α/β-hydrolase core domain (blue) of mCes2c shows differences to hCES1 (wheat). (**a**–**d**): The three catalytic triad residues are shown as green sticks.

**Table 1 ijms-23-13101-t001:** Statistics of X-ray data collection and structure refinement.

Contents	mCes2c
**PDB code**	**8AXC**
**Data collection**	
Data collection facility	Synchrotron, PETRA III, DESY BEAMLINE P11 Hamburg
Detector	DECTRIS EIGER2 X 16M
Wavelength (Å)	1.0
Resolution range (low–high Å)	49–2.12 (2.19–2.12)
Space group	P2_1_2_1_2_1_
No. of molecules in the asymmetric unit	4
Unit cell (Å, °)	a = 98.05 b = 143.59 c = 183.75 α = β = γ = 90
Total reflections	1,006,260 (89,693)
Unique reflections	145,761 (14,158)
Multiplicity	6.9 (6.3)
Completeness (%)	98.99 (97.40)
Mean I/σ	16.58 (3.67)
CC1/2	0.99 (0.87)
**Refinement statistics**	
R-merge	0.090 (0.54)
Reflections used in refinement	145,744 (14,158)
Reflections used for R-free	7342 (711)
R-work	0.166 (0.213)
R-free	0.195 (0.242)
**Number of non-hydrogen atoms**	18,010
Macromolecules	16,430
Ligands	63
Water atoms	1517
No. of protein residues	2069
**Model geometry**	
RMSD bonds (Å)	0.014
RMSD angles (°)	1.75
**Ramachandran distribution**	
Most favored (%)	97.27
Additionally allowed (%)	2.73
Outliers (%)	0
**Average B-factor (A^2^)**	35.23
Macromolecules	34.92
Ligands	37.41
Solvent	38.43

Values in parentheses are for the highest resolution shell. Crystal parameters and data collection statistics are derived from Phenix.table_one.

## Data Availability

Experimental data for determining the structure of mouse Ces2c have been deposited in the PDB (PDB-ID 8AXC).

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
