# Peer review of "The Crystal Structure of Mouse Ces2c, a Potential Ortholog of Human CES2, Shows Structural Similarities in Substrate Regulation and Product Release to Human CES1"

_ijms, 2022, doi:10.3390/ijms232113101_

Round 1

Reviewer 1 Report

The authors report the crystal structure of mouse carboxylesterase 2c and biophysical analyses to show that the protein behaves as a monomer in solution. The structure of ces2c is compared to human CES1 and similarities and differences carefully described and features that may account  for the differing substrate selectivity of these two types of carboxylesterase are identified. Mouse ces2 enzymes appear to be homologues of the single human CES2 enzyme and therefore this study provides a very useful model for understanding the structure-activity of human CES2 which is the key esterase involved in Phase I metabolism in GI tract. As a minor comment it would be helpful to include an annotated  sequence alignment of mces2c and hCES1 and 2 e.g. with active site residues conserved disulphides. 

Author Response

We, the authors, thank reviewer 1 for the very positive evaluation of our manuscript and the constructive comment which has been implemented in the revised version. Furthermore, we corrected minor spelling errors or grammatical mistakes throughout the manuscript to further improve readability. The detailed answer to reviewer 1 is listed below.

Reviewer 1:

As a minor comment it would be helpful to include an annotated  sequence alignment of mces2c and hCES1 and 2 e.g. with active site residues conserved disulphides. 

Answer:

According to the reviewer’s suggestion, we included an annotated alignment of mCes2c, hCES2, and hCES1 in the supplementary material (Supplementary Figure 1) and referred to the alignment in the results section and in the discussion section.  

Reviewer 2 Report

In “The crystal structure of mouse Ces2c, a potential ortholog of human CES2, shows structural similarities in substrate regulation and product release to human CES1”, Eisner and colleagues describe the molecular characteristic feature of mouse Ces2c, a potential ortholog of human CES2.

In my opinion, the manuscript can be published after a major revision, where authors revise their manuscript to address significant concerns.

Major points

About “NCA, nicotinamide”

It is very important to show the electron density map of NCA, is only method to show the evidence of binding NCA molecule.

Thus, the author should show the electron density map of NCA.

The author should check why the chain C and D dose not bind NCA.

For example, in chain C and chain D, because of the crystal packing, the active site of chain C and D cover by the neighbor molecule.

The author should discuss the binding manner of NCA.

For example, show the residues around 4â„« of NCA and hydrogen bond to the amino acid residues of the Ces2c.

It is better to show the chemical structure of NCA.

Minor points

In figure 1 (a), it is better to show the residue number of Ces2c between the regions not only catalytic residues.

In table 1, the author should divide the column of “No. of protein atoms” to “No. of protein atoms, NCA atoms and water atoms”.

In table 1, the author should divide the column of “Average B-factor” to “protein, NCA and water”.

In figure 4 (or text in gel filtration chromatography part), the author should mention about the protein containing the gel filtration standard from BioRad.

For example, thyroglobulin (bovine); 670 kDa, g-globule (bovine); 158 kDa, ovalbumin (chicken); 44kDa, myoglobin (horse); 17 kDa, vitamin B12; 1.35 kDa 

In the figure 4(a), it is better to show the molecular weight in the chromatogram of the molecular weight maker gel filtration.

Author Response

Answer:

We, the authors, also thank reviewer 2 for the very positive evaluation of our manuscript and the constructive comments which have been implemented in the revised version. Furthermore, we corrected minor spelling errors or grammatical mistakes throughout the manuscript to further improve readability. The detailed answer to reviewer 2 is listed below.

  • About “NCA, nicotinamide”. It is very important to show the electron density map of NCA, is only method to show the evidence of binding NCA molecule. Thus, the author should show the electron density map of NCA. The author should check why the chain C and D dose not bind NCA. For example, in chain C and chain D, because of the crystal packing, the active site of chain C and D cover by the neighbor molecule. The author should discuss the binding manner of NCA. For example, show the residues around 4â„« of NCA and hydrogen bond to the amino acid residues of the Ces2c. It is better to show the chemical structure of NCA.

We are grateful for the comment to include more information on the ligand nicotinamide (NCA). For the revised version, we have included the electron density maps of NCAs in the different sites and show the chemical structure of NCA in the electron density (Supplementary Figure 2). The panels include the electron density of NCA modelled into the active site of mCes2c in chains A-D. Furthermore, we included a LigPlot+ image, demonstrating the predominant interactions of the carbonyl-oxygen from NCA with the backbone amides of the oxyanion-hole residues G151, A152, A231, and H-bonding with a water molecule.

The corresponding section now reads: “In each monomer of the experimental structure of mCes2c, additional electron density was observed that could ideally be fitted to one molecule of nicotinamide (NCA) in the substrate binding pockets with 100% occupancy (Supplementary Figure 2). Analysis of the ligand-protein interaction reveals H-bonding of the NCA-carbonyl oxygen to the backbone amide protons of oxyanion-hole residues G151, A152, and A231. Additionally, H-bonding to a modelled water molecule is observed (Supplementary Figure 2).”

The electron densities of the ligand in the Z-site of chains A-D are displayed in Supplementary Figure 3. In the deposited structure, NCA at the Z-site of chain A is placed with 70% occupancy, at the Z-Site of chain B with 82% occupancy. Some residual density is present in chains C and D. We do not see an obvious reason, why the sites are not fully occupied, even after checking crystal packing. References pointing towards the new supplementary figures have been included, and a sentence describing the interactions of NCA in the active site with the protein has been added in the revised version of the manuscript (section 2.4). The statement now reads: “In the structure we found electron density at the Z-site and could fit a single NCA molecule in chains A and B. Only residual electron density can be observed in chains C and D, which could not be fitted unambiguously to a distinct small molecule with full occupancy (Figure 3b, Supplementary Figure 3).”

  • In figure 1 (a), it is better to show the residue number of Ces2c between the regions not only catalytic residues.

We included the residue numbers between the different regions of mCes2c. Furthermore, we also included an annotated and color-coded alignment of mCes2c, hCES2, and hCES1 in Suppl. Figure 1 as suggested by Reviewer 1.

  • In table 1, the author should divide the column of “No. of protein atoms” to “No. of protein atoms, NCA atoms and water atoms”.
  • In table 1, the author should divide the column of “Average B-factor” to “protein, NCA and water”.

The changes regarding Table 1 have been included in the revised version of the manuscript.

  • In figure 4 (or text in gel filtration chromatography part), the author should mention about the protein containing the gel filtration standard from BioRad. For example, thyroglobulin (bovine); 670 kDa, g-globule (bovine); 158 kDa, ovalbumin (chicken); 44kDa, myoglobin (horse); 17 kDa, vitamin B12; 1.35 kDa .In the figure 4(a), it is better to show the molecular weight in the chromatogram of the molecular weight maker gel filtration.

The changes regarding Figure 4a have been implemented in the revised version: We included the molecular weights directly in the gray chromatogram of the standard and included the molecular weights of the standards along with the protein names in the figure legend. 

Reviewer 3 Report

Figure 6. HIP459 residue - Should it be HIS459?

In Methods section, full length mCes2c was used. However mCes2c was expressed in Expi293F in secreted form. Was secretion signal sequence added to the construct? If yes, please add more detail to the Methods section.

Author Response

Reviewer 3:

We, thank reviewer 3 for the very positive evaluation of our manuscript and the constructive comments which have been implemented in the revised version. We also corrected minor spelling errors or grammatical mistakes throughout the manuscript to further improve readability. The detailed answer to reviewer 3 is listed below.

  • Figure 6. HIP459 residue - Should it be HIS459?

We apologize for the lack of clarity in the figure description. HIP459 refers to the doubly protonated state of H459. The protonation-state of histidine residues is estimated by the docking program at the H-bond optimization stage when the chemical environment in a 3.5 Å radius of H459 is taken into consideration. To avoid confusion, we put ‘HIS459*’ in Figure 6a of the revised version. The figure legend now reads: “Molecular docking of the prodrug molnupiravir into mCes2c. (a) The ligand interaction diagram shows mCes2c residues within a 3.5 Å radius of molnupiravir. Pink arrow denotes H-bonding interaction. The asterisk * is placed to indicate that H459 is in the doubly protonated state. (b) 3D image displaying the proximity of active site residues to the ligand. The distance from carbonyl carbon of molnupiravir and the oxygen of S230 is 3.4 Å. The catalytic triad and oxyanion hole residues are depicted as green and blue sticks, respectively”

  • In Methods section, full length mCes2c was used. However mCes2c was expressed in Expi293F in secreted form. Was secretion signal sequence added to the construct? If yes, please add more detail to the Methods section.

The coding sequence for mCes2c has been cloned into the multiple cloning site of the vector pFLAG-CMV™-5.1 using restriction sites EcoRI and XbaI (see Chaloub et al, 2021;  doi: 10.1016/j.jlr.2021.100075). No further secretion signal was added to our construct, yet the protein is still secreted into the medium. We suggest that the secretion of the intact protein is due to high overexpression of mCes2c. No changes regarding the secretion of mCes2c were thus included in the revised version.

Round 2

Reviewer 2 Report

I read the answers for referee’s questions and new paper.

The paper changed for the referee’s comments and I think that it is worthy for publishing.

 Yours sincerely,